# Two New Cases of Bachmann–Bupp Syndrome Identified through the International Center for Polyamine Disorders

**DOI:** 10.3390/medsci11020029

**Published:** 2023-04-04

**Authors:** Julianne Michael, Elizabeth VanSickle, Marlie Vipond, Abby Dalman, Jeremy Prokop, Charles E. Schwartz, Surender Rajasekaran, André S. Bachmann, Magalie Barth, Clément Prouteau, Yotam Almagor, Lina Berkun, Gheona Alterescu, Caleb P. Bupp

**Affiliations:** 1Corewell Health and Helen DeVos Children’s Hospital, Grand Rapids, MI 49503, USA; 2International Center for Polyamine Disorders, Grand Rapids, MI 49503, USA; 3Department of Pediatrics and Human Development, College of Human Medicine, Michigan State University, Grand Rapids, MI 49503, USA; 4Department of Biochemistry and Genetics, Angers University Hospital Center, 49100 Angers, France; 5Meuhedet Healthcare, Jerusalem 9957520, Israel; 6Shaare Zedek Medical Center, Medical Genetics Institute, Jerusalem 9103102, Israel

**Keywords:** Bachmann–Bupp Syndrome (BABS), Snyder–Robinson Syndrome (SRS), International Center for Polyamine Disorders (ICPD), α-difluoromethylornithine (DFMO), polyamine

## Abstract

Recent identification of four additional polyaminopathies, including Bachmann–Bupp syndrome, have benefited from previous research on Snyder–Robinson syndrome in order to advance from research to treatment more quickly. As a result of the discovery of these conditions, the potential for treatment within this pathway, and for other possible unidentified polyaminopathies, the International Center for Polyamine Disorders (ICPD) was created to help promote understanding of these conditions, research opportunities, and appropriate care for families. This case study provides insights from two new patients diagnosed with Bachmann–Bupp syndrome, further expanding our understanding of this ultra-rare condition, as well as a general discussion about other known polyaminopathies. This work also presents considerations for collaborative research efforts across these conditions, along with others that are likely to be identified in time, and outlines the role that the ICPD hopes to fill as more patients with these polyaminopathies continue to be identified and diagnosed.

## 1. Introduction/Background

Snyder–Robinson syndrome (SRS) OMIM #309583, a neurocognitive condition notable for intellectual disability as well as bone and muscle issues, was first described clinically in 1969 [1] and the causative gene, *SMS*, was identified in 2003 [2]. Since then, the question of genetic syndromes caused by variants in other genes in the polyamine pathway has essentially gone unanswered (Figure 1). In 2018, the second polyamine pathway condition, Bachmann–Bupp syndrome (BABS) OMIM 619075, was identified by pathogenic variants in the C-terminus of the *ODC1* gene [3,4]. Primary dermal fibroblasts from a patient with BABS were shown to contain large amounts of ornithine decarboxylase (ODC) protein and putrescine [5]. The hallmarks of this condition are developmental delay, hypotonia, non-specific brain MRI findings, non-specific dysmorphic features, and non-congenital alopecia. More specifically, patients are born with hair, which is sometimes sparse and atypical in color, that falls out in clumps in the first few weeks of life. Some individuals have regrowth of scalp hair that typically remains sparse, with congenital absence of the eyebrows and eyelashes. Additionally, though non-specifically, 7/10 patients so far have had polyhydramnios prenatally. Four additional syndromes caused by pathogenic variants in genes in the polyamine pathway have been described (Figure 1). There are at least seventeen potential additional conditions in the polyamine pathway that are yet unknown, a fact which led to the formation of the International Center for Polyamine Disorders (ICPD). Located in Grand Rapids, Michigan, the ICPD exists for comprehensive data generation and local as well as remote sample collection from patients with known or suspected polyamine disorders around the world. The ultimate goal of the ICPD is to advance scientific knowledge of polyamine-associated genetic disorders and medical care for families impacted by these conditions. The ICPD is a collaboration between Corewell Health and Michigan State University, with strong involvement from the Snyder-Robinson Foundation.

In this update, information on two additional patients with BABS is presented, the other polyamine pathway conditions are briefly described, and phenotypic themes and treatment possibilities are examined.

## 2. Additional BABS Case Presentations

### 2.1. Patient 10

This patient is a 12-year-old Caucasian-South Pacific female at the time of reporting and was born at 39 weeks gestation after a pregnancy complicated by polyhydramnios and maternal pre-eclampsia. At birth, she measured at the 57th percentile (49.5 cm) for height, 85th percentile (3735 g) for weight, and greater than the 99th percentile (38 cm) for head circumference, based on World Health Organization (WHO) 0–2 years standard growth curves. An examination at age 12 identified her height as being between the 25th and 50th percentiles and her weight as being at the 50th percentile, based on the Centers for Disease Control and Prevention (CDC) 2–18 years standard growth curves. At age 7, her head circumference was between the 96th and 97th percentiles for CDC 2–18 years standard growth curves. She walked independently at 4 years of age with physiotherapy and had noted delays in standing and sitting independently. She has had an increase in hypotonia over time and has developed proximal myopathy identified by electromyogram EMG, bilateral medius lameness, and bilateral heel defects leading to walking degradation and the need of a wheelchair for long distances along with difficulty climbing stairs. At the age of 2 years and 1 month, she was using five words. At 12 years of age, she has moderate developmental delay and intellectual disability, with the ability to read and write. Behaviorally, she is noted to be hypersensible and have frustration intolerance. A brain MRI identified myelination impairment with dilated Virchow–Robin spaces, bilateral paraventricular cysts, enlargement of subarachnoid spaces associated with ventricular dilatation, and hypoplasia of the middle part of the corpus callosum. Additionally, after renal imaging, she was shown to have a left bifid malformed kidney. An echocardiogram yielded unremarkable results. Her past medical history also includes hyperlaxity and anhidrosis with hyperthermia episodes. Dysmorphic facial features include thin hair and the absence of eyebrows and eyelashes.

She has a history of previous nondiagnostic Noonan disorder panel and cognitive disorders panel. After trio exome sequencing, she was diagnosed as having a de novo heterozygous variant in the ODC1 gene, c.1242-2A > G, (IVS11-2A > G), which is predicted to destroy a canonical splice acceptor site in intron 11. This variant is not observed in the population [6,7,8]. Her providers reached out to Dr. Bupp, who connected the ICPD team to the family.

### 2.2. Patient 11

This patient is a 12-month-old female at the time of reporting and was born at term. Concerning prenatal findings included cerebral ventricular enlargement, cerebral cysts, polyhydramnios, and a ventricular septal defect. At birth, she measured at the 10th percentile (2750 g) for weight, based on WHO 0–2 years standard growth curves. Postnatally, she exhibited respiratory distress, requiring oxygen and NICU stay for four weeks prior to discharge on oxygen. A brain MRI identified cerebral ventricular enlargement, right subependymal cyst, and vascular calcifications. Her past medical history also includes hypotonia, feeding difficulties, food aspiration, marked joint hypermobility, and hepatic calcifications. Dysmorphic facial features include essentially no eyelashes, few faint eyebrows upon close inspection, scalp alopecia outside of tuft of long and coarse hair on central posterior scalp, erythematous vascular marking on posterior head, and elongated head with sloping forehead. At the time of reporting, the patient is receiving physical, occupational, and speech therapy services. She can sit upright with minimal assistance, self-feed independently, and primary mobility is achieved through rolling and compensated commando crawling.

After trio whole genome sequencing, she was diagnosed as having a de novo heterozygous variant in the ODC1 gene, c.1307_1311delinsT (p.Thr436IlefsX11), which is predicted to shift the reading frame, resulting in premature truncation of the protein on genome sequencing. This variant is not observed in the population [6,7,8]. Her providers reached out to Dr. Bupp, who connected the ICPD team to the family.

## 3. Discussion

BABS was the second identified genetic syndrome related to the polyamine pathway, with eleven patients now reported. Common features seen among these individuals are an unusual pattern of non-congenital alopecia, significant developmental delay, macrocephaly, non-specific dysmorphic features, hypotonia, non-specific neuroimaging differences, and prenatal polyhydramnios as noted in Table 1 [9]. The two patients reported here further expand the understanding of this condition by reiterating these common features as well as bringing to light potential new associated phenotypes. Specifically, Patient 10 is the first reported with a history of progressive hypotonia and proximal myopathy, though a majority of the patients reported are younger than this patient and therefore may not have developed these concerns yet. Notably, this patient is also the third BABS patient reported with the same intronic finding, which raises the notion of potential hotspots within the gene. Patient 11 had more significant immediate post-natal concerns than the majority of the patients so far reported and is also the youngest patient to begin previously published research treatment with the ODC inhibitor α-difluoromethylornithine (DFMO).

After SRS and BABS, three other polyamine pathway conditions have now been described (Figure 1). The BABS experience of identifying the syndrome, refining the clinical features, and selecting the treatments showcases how related syndromes can inform each other.

The first description of SRS came after a clinical observation was made of multiple affected males in a large extended family in an apparent sex-linked pattern. Intellectual disability, hypotonia, and unsteady gait were the initial hallmarks of the condition [1]. Only over time was the genetic cause (spermine synthase (*SMS)* gene on the X chromosome) found [2], and additional characteristic features noted and clarified. Based on the biochemical pathway impacted by SRS, treatment options have been proposed and tested, including spermine supplementation, prodrug treatment, and drug repurposing [10,11,12]. The presence of *ODC1* and its known inhibitor, DFMO, close to *SMS* in the polyamine pathway provided some general background upon which to consider research treatment quickly when BABS was discovered.

Drug repurposing creates opportunities for rapid treatment in certain circumstances. In the situation of BABS, three main factors played a role in moving from publication of the syndrome to research treatment in the first patient in only 15 months. First, the repurposed drug DFMO had been synthesized and used for over 40 years for a variety of indications. When considering its use in pediatric treatments for BABS, its use in pediatric patients for neuroblastoma provided valuable usage and safety data [13,14]. Second, a conditional transgenic mouse model that constitutively overexpresses ODC1 with a premature stop codon, resulting in a C-terminal truncation, had been created in 1996 which matched the human skin phenotype. When those mice were treated with DFMO, their hair regrew [15,16]. Lastly, primary cells from a BABS patient treated with DFMO showed promising results [5]. Results in the patient from this treatment showed both clinical and biochemical improvement [17].

Since BABS was published in late 2018 [3], syndromes caused by changes in other polyamine pathway genes, deoxyhypusin synthase (*DHPS)* in early 2019, eukaryotic translation initiation factor 5A (*eIF5A)* in 2021, and deoxyhypusine hydroxylase (*DOHH)* in 2022, were all described [18,19,20,21]. Differing models and evidence were used in each, including yeast, zebrafish, and mice, as well as enzyme analysis and biochemical analysis. Similar to SRS and BABS, all have neurodevelopmental phenotypes and delays, but not consistent physical features between them. All appear to be very rare with few reported patients. Growth differences (head size and length) in all polyamine-related syndromes seem to be features that regularly, but not consistently, exemplify the question of what the cardinal features are of rare disorders when only a few patients have been reported.

The presence of now five defined genetic syndromes in the polyamine pathway is notable for three reasons. The first is that the causative gene for SRS was reported in 2003 (linkage suggested location in 1996) while the other four were identified in a span of 4 years [2,18,19,20,21,22]. This speaks to the rapid growth in genetic testing, novel gene identification, and matchmaking available worldwide. Second, while there are five now known, the polyamine pathway consists of over fifteen additional genes with no currently known condition (Figure 1). In time, and perhaps in the near future, more will likely be discovered. Third, the biochemical pathway implicated in these conditions presents many opportunities for treatment, let alone multiple conditions potentially treated with the same drug or combination of drugs. The question of how to reference these conditions can be raised to allow for a better global understanding of their commonalities and differences, which could help advance understanding, diagnosis, and treatment of one or more of the included conditions. For this reason, we suggest referring to these conditions as polyaminopathies, and propose that a more in-depth literature review of these conditions would likely be of benefit.

Unifying these conditions also helps bring into focus the challenges that are faced by patients, clinicians, and researchers. The ultra-rarity of the conditions is certainly a barrier. Even the most well-known and studied of the polyaminopathies, SRS, is only known to affect 80–100 individuals worldwide. To date, most polyaminopathies have been identified by molecular testing, which introduces the opportunity for variants of uncertain significance that differ in their ability to resolve. This is particularly challenging for *ODC1* variants found outside the C-terminus region that may have a differing impact on protein function. There is no standardized clinical biochemical testing for polyamine levels and ratios available commercially, though HPLC methods to determine polyamine ratios and global metabolomics analysis may help. DNA methylation signatures have already been shown to exist for SRS and could be present in other polyaminopathies [23]. At present, some patients and families are still dealing with uncertain diagnoses of polyaminopathies, which is increasingly distressing as treatment options begin to become available.

In reaction to the opportunities and challenges surrounding polyaminopathies, the ICPD was created in 2020 to combine clinical experts in rare disease genetics with expert bench scientists studying polyamines in order to help advance and support global polyamine research and clinical care. To facilitate this, a biobank of high-quality serial biological samples from patients with known and suspected polyamine disorders as well as their full biological family members was established. Samples from patients with SRS from the Greenwood Genetic Center in Greenwood, South Carolina (where the *SMS* gene was identified), have been received into the biobank as well as samples from additional patients with SRS who attended the 2022 Snyder-Robinson Foundation Conference in Grand Rapids, Michigan. Additionally, the biobank continues to expand to include samples from other patients with polyamine disorders identified from around the world. The ICPD collaborates with many leading academic and industry partners to achieve cutting-edge transcriptomic and metabolomic data interrogation and analysis to better understand the landscape of these ultra-rare disorders.

## 4. Conclusions

The addition of two newly identified BABS patients to a total of eleven reported further strengthens the phenotype of this ultra-rare disorder. Given the recent description of other rare disorders in the polyamine pathway (DHPS, eIF5A, and DOHH)—all described in the literature since 2019, it is likely that other conditions within this pathway will be identified as molecular testing capabilities expand. As a result, the ICPD continues to expand its collaborative research efforts to meet the needs of the growing community of patients with polyaminopathies.

Despite recent advances in clinical care and research, challenges within this family of disorders remain. Since no established clinical testing exists for measuring polyamine levels, or, in the case of BABS, ODC activity, diagnosing patients with these ultra-rare disorders remains challenging for those unfamiliar with polyaminopathies and for those families that receive a variant of uncertain significance in a gene associated with a polyaminopathy. Additionally, treatment of these rare diseases is challenging and typically involves research into repurposing FDA-approved drugs using the pediatric safety data available (as is the case for DFMO). While less costly and time-consuming than drug development, efforts to obtain Investigational New Drug (IND) approval are limited by the availability and willingness of pharmaceutical companies to supply the drug for all patients seeking treatment.

## Figures and Tables

**Figure 1 medsci-11-00029-f001:**
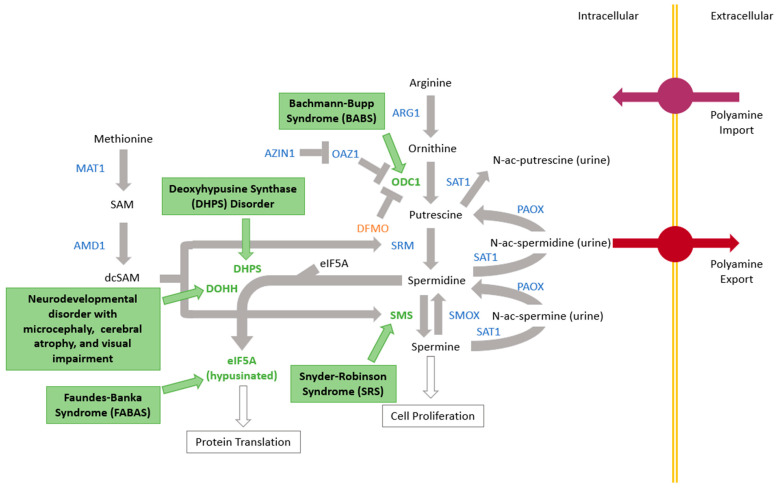
Polyamine pathway with five known genetic syndromes shown in green text boxes by the gene associated. Other pathway proteins are denoted in blue.

**Table 1 medsci-11-00029-t001:** Patient genotypes and phenotypes.

	Patient 10	Patient 11	Total
Variant NM_001287190.1	c.1242-2 > G (p.IVS11-2A > G)	c.1307_3111delinsT (p.Thr436Ilefs*11)	--
Inheritance	De novo	De novo	11/11 de novo
Age at most recent evaluation	12 years	2 months	--
Sex	Female	Female	6/11 male 5/11 female
Prenatal findings	Polyhydramnios, maternal pre-eclampsia	Cerebral ventricular enlargement, cerebral cysts, polyhydramnios, ventricular septal defect	7/10 polyhydramnios
Head circumference	95–97% at age 7	0.1% (31.5 cm) at 1 month	--
Height	57%	86% (61 cm)	--
Weight	85%	99% (7711 g)	--
Global developmental delay	Yes	Yes	10/10
Age at walking	4 years	Not yet	--
Age at first words	Unknown—speaking 5 words at 2 years 1 month	Not yet	--
Behavior	Hypersensible, frustration intolerance	--	6/8
Epilepsy	No	No	1/10
Hypotonia	Yes	Yes	9/10
Dysmorphic features	None	Elongated head with sloping forehead	8/11 forehead differences 6/11 hypertelorism
Skin	Normal	Erythematous vascular marking on posterior head	--
Hair	Thin hair, absence of eyebrows and eyelashes	Essentially no eyelashes, few faint eyebrows upon close inspection, scalp alopecia except for tuft of long and coarse hair on central posterior scalp	9/11 partial alopecia
Other	Proximal myopathy	Joint hypermobility, food aspiration, hepatic calcifications	--
MRI, brain	Myelination impairment with dilated Virchow–robin spaces, bilateral paraventricular cysts, enlargement of subarachnoid spaced associated with ventricular dilatation, corpus callosum hypoplasia	Cerebral ventricular enlargement, right subependymal cyst, vascular calcifications	--

## Data Availability

No new data were created or analyzed in this study. Data sharing is not applicable to this article.

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
