# Peer review of "Two New Cases of Bachmann–Bupp Syndrome Identified through the International Center for Polyamine Disorders"

_medsci, 2023, doi:10.3390/medsci11020029_

Round 1

Reviewer 1 Report

The manuscript by Michael et al. is a case report that describes two new patients diagnosed with Bachmann-Bupp Syndrome (BABS). This is an extremely rare disorder caused by heterozygous mutations in the ornithine decarboxylase 1 (ODC1) and only nine cases were reported prior to this manuscript. In addition to the case report, the authors briefly describe the genetic syndromes caused by variants of other polyamine genes. Finally, the authors introduce International Center for Polyamine Disorders (ICPD) in this manuscript. ICPD was created for comprehensive data generation and sample collection from patients with the polyamine disorders and their goal is to understand these rare genetic syndromes, advance polyamine research and improve medical care for the patients. 

Overall, the manuscript is well written and describes the patient conditions in detail, which further expand our knowledge of this syndrome. These two cases show previously described common features including development delay, hypotonia, and alopecia as well as some new features. To understand a rare genetic disease like BABS, it is critical to continue to collect data from the new cases. ICPD is trying to establish a global comprehensive sample/data of polyamine-associated genetic disorders including BABS in collaboration with other academic and industry partners. There is no doubt that such collaborative approach would better identify the cases and accelerate the research for polyamine disorders to develop new treatment options as well as the molecular diagnostic testing of these disorders. As such, I recommend publishing of this case report in its present form in Medical Sciences.

Author Response

Reviewer 1 recommended publication of the manuscript as it stands with no additional edits.

Reviewer 2 Report

The authors are experts in polyamine-related disorders and foundational to identifying patient cases as described in this update. The article comes timely and describes the clinical parameters of the identified patients 10 and 11 with mutations in the ODC gene. It is a well-written and informative piece, highly relevant to the specific field and beyond.

There is not much to criticize, but I have a few questions/points:

- How exactly were the patients identified? There is no mention of the screening and/or sequencing procedure.

- The introduction focuses mainly on the ICPD. It would be good to have some more background identification about the previously identified patients and the genetic, as well as biochemical, features of the disease.

- It would be a great added value to the paper if the authors could determine systemic or tissue-specific polyamine levels in those patients.

- While the ICPD is clearly an important institution, the first sentence of the conclusion is not needed in this case report, in my personal opinion.
